# Iridescent Features Correlating with Periodic Assemblies in Custom-Crystallized Arylate Polyesters

**DOI:** 10.3390/ijms242115538

**Published:** 2023-10-24

**Authors:** Widyantari Rahmayanti, Selvaraj Nagarajan, Ya-Sen Sun, Eamor M. Woo

**Affiliations:** Department of Chemical Engineering, National Cheng Kung University, No. 1, University Road, Tainan 701-01, Taiwan; widyantarii32@gmail.com (W.R.); nagarajan.tech@gmail.com (S.N.)

**Keywords:** aryl polyesters, ring-banded spherulites, crystal morphology, iridescent properties

## Abstract

In this study, five different aryl polyesters, i.e., poly(ethylene terephthalate) (PET), poly(trimethylene terephthalate) (PTT), poly(octamethylene terephthalate) (POT), poly(nonamethylene terephthalate) (PNT), and poly(decamethylene terephthalate) (PDT), upon crystallization at a suitable temperature range, all exhibit ring-banded spherulites with universal characteristics. Previous research has revealed some fundamental mechanisms underlying the formation of periodic hierarchical structures. Additionally, this study further explored correlations among micro/nanocrystal assemblies in the top surface and internal grating architectures and the structural iridescent properties. The interior lamellar assembly of arylate polyesters’ banded spherulites is shown to exhibit periodic birefringence patterns that are highly reminiscent of those found in a variety of biological structures, with the capacity for iridescence from light interference. A laser diffraction analysis was also used to support confirmation of this condition, which could result in an arc diffraction pattern indicative of the presence of ringed spherulites. Among the five arylate polyesters, only PET is incapable of regularly producing ring-banded morphology, and thus cannot produce any iridescent color.

## 1. Introduction

Crystalline aggregates form in a large hierarchical order; in polymer crystallization, this can be achieved under conditions of single crystal formation (thin film, composition of the solution, crystallization temperature, etc.) [1,2,3,4,5,6,7,8,9]. The growth of crystalline aggregates is not confined to polymers; it can also occur in tiny molecule compounds with similar optical patterns [10,11,12,13,14]. For the past half a century, the phenomena of circular ringed spherulites with periodicity and optical birefringence spiral/concentric bands have attracted interest [15,16,17,18,19]. Polymer spherulites are composed of lamellae-like crystals that are self-arranged by radiating outward from a common center; often, polymer spherulites display periodic bands, as exemplified in several recent review articles [20,21,22,23,24,25,26]. Such a ring-banded assembly is also common in aromatic polyesters, which are defined by repeating chemical units of varying lengths of methylene segments between two terephthalate groups. Arylate polyesters are known to possess a number of methylene segments with unit numbers ranging from 2 to 20. Three commercially useful aryl polyesters are poly(ethylene terephthalate) (PET), poly(trimethylene terephthalate) (PTT), and poly(butylene terephthalate) (PBT) with repeated terephthalate segments 2, 3, and 4, respectively. Other aryl polyesters can be produced synthetically, such as poly(octamethylene terephthalate) (POT) and poly(decamethylene terephthalate) (PDT) [27,28].

Except for PET, PBT, and poly(hexamethylene terephthalate) (PHT), most other aryl polyesters, like poly(trimethylene terephthalate) (PTT) [29,30,31], poly(octamethylene terephthalate) (POT) [32], and poly(nonamethylene terephthalate) (PNT) [33] are known to display ring-banded spherulitic morphology upon melt crystallization at certain T_c_ values. Although some exceptions have been noted, melt-crystallized PET with well-ordered ring-banded spherulites have rarely been reported, although solution-cast PET films might exhibit irregularly ordered ring bands. On the one hand, PET has a triclinic crystal cell with dimensions a = 4.56 Å, b = 5.94 Å, c = 10.75 Å, α = 9.5°, β = 118°, and γ = 112°; furthermore, it does not possess crystal-lattice polymorphism [28,34]. PTT, on the other hand, easily forms a periodic assembly. Lugito et al. [35] observed three types of nuclei at the same crystallization temperature in PTT. Different ring-shaped spherulites are clockwise double-spiral ring-banded spherulites, single-spiral ring-banded spherulites, and concentric ring-banded spherulites, in which the nuclei’s positions and geometry determine the growth of spherulites. Chen et al. [36] reported a phenomenon, in 2008, on the growth kinetics and morphology of POT spherulites. POT can exhibit a single or two types of spherulites depending on the crystallization temperature or melting temperature under various conditions. In 2012, Woo et al. [37] used AFM to investigate the relatively rare dual types of banded spherulites in PNT crystallized at the same T_c_. Each type had a different proportion according to the crystallization temperature. The band spacing, POM birefringence, surface topography, and interior lamellar assembly of each of the two types of ring bands were significantly different. In 2021, Yang et al. [38] probed the detailed mechanism of the lamellar assembly in neat PDT. By crystallizing neat PDT at two different T_c_ ranges, all the spherulites exhibited double ring-banding at lower Tc values (80–110 °C) and higher T_c_ values (110–115 °C). However, as T_c_ rises above 115 °C, the morphology changes to an epicycloid extinction-ring band. Within this scope, this work concentrates on the outcomes of a series of aryl polyesters (with terephthalate groups) instead of addressing the enormous variety of polymers structures. The polymers with banded spherulites were used to study the formation of crystals on aliphatic polyesters during polymerization and crystal assembly. As a result, optically ringed polymer spherulites are formed, such as poly(ethylene adipate) (PEA), which is one of the most widely studied polymers [39]. 

Orderly arrays in microstructures might come with periodic assemblies in polymer crystals, which potentially are capable of interference with optical white light into spectral coloration. Starting about 60 years ago, a new discovery sparked the paradigm for the formation of natural micro/nanostructures found in living species that are responsible for creating coloration [40]. Color-producing architectures have been confirmed to occur not so uncommonly in animals, insects, plants, or inorganic minerals when exposed to white light or ultraviolet (UV) light. This has been clearly observed in the fruit skins of Margaritaria nobilis [41], wings of Papilio blumei butterfly [42], and the nacre of Hyriopsis cumingii [43]. Photonic phenomena have been obtained through in-depth analyses of top surfaces and cross-sections. According to the arrangement of lamellae on the surface and inner surface topography, polymers also have grating assemblies similar to nature’s structural crystals due to the orderly arrangement of the lamellae in their periodic crystal aggregates.

Many polymeric materials with banding patterns have different topologies at the surface, which correlate inherently to variations in interior structures. The varieties of ring bands were evaluated, as well as their variation tendencies. Furthermore, this work aimed to compare the iridescent properties of a series of five different aryl polyesters, namely PET, PTT, POT, PNT, and PDT. Aryl polyesters with their unique ring-banded morphologies were chosen as ideal models to interpret the iridescent properties from the hidden microstructures in the diversified ring-banded morphologies. Systematic investigations were conducted by adjusting the thermal treatments, crystallization parameters, etc., of the aryl polyesters. Moreover, through this hierarchical structure of aryl polyesters, the photonic iridescent phenomena of the orderly structures in crystallized polymers could be used as critical supporting evidence for the microarrays in the aggregated crystals. A reasonable mechanism for the light interference phenomena and a correlation with the polymer iridescent properties have been established through detailed investigation. Understanding the correlation between different morphologies, crystallization behaviors, and iridescent properties of aryl polyesters would be incredibly beneficial for their future use in advanced applications, such as interference filters [44], currency anti-counterfeiting features [45], and security documents [46]. In the realm of materials science, it is used to create specialized pigments for paints, cosmetics, and coatings to enhance their visual appeal [41,47,48,49,50,51,52,53,54]. This study of iridescence also has implications for biomimicry, inspiring innovations in materials and coatings across diverse industries. 

## 2. Results and Discussion

Five different aryl polyesters, including PET and PTT, were investigated and compared. By comparison, PTT, which contains one more methylene unit than PET in the repeat units, and three other aryl polyesters containing longer methylene segments, such as POT, PNT and PDT, all exhibit ring-banded morphology. Note that poly(butylene terephthalate) (PBT) and poly(hexamethylene terephthalate) (PHT) are not included in the investigation list, because these two arylate polyesters are not able to display any ring bands when crystallized at all T_c_s. Unlike the commercialized PET, PTT, and PBT, the aryl polyesters with longer methylene segments (POT, PNT, and PDT) have received less attention due to their limited application potentials. However, their common ability to generate a periodically ring-banded morphology is scientifically intriguing and should be investigated further. Considering these facts, one may speculate that these periodically banded aryl polyesters might have micro/nanostructures to resemble nature’s photonic crystals in organic or inorganic species. To address the intellectually probing questions, this study clarified the correlation between the crystal morphology and iridescent properties in this study; furthermore, iridescent tests were performed on the polyarylate films, crystallized at a suitable temperature to generate orderly rings, to confirm the interior lamellar assembly, further justified with the capacity for interference with light.

### 2.1. Morphology and Iridescent Properties of PET and PTT

PET is known to be incapable of producing ring-banded spherulites at all crystallization temperatures and can be regarded as “hardly” ringed spherulites. Although ring-banded spherulites have been reported infrequently in PET, it has been demonstrated that some exceptions exist, with solution-cast PET films exhibiting an irregular ring-banded pattern. Supaphol et al. [55] studied PTT and PET blend’s thermal properties, and the results suggested that PTT was more crystallizable than PET. To begin the discussion on these diversified aryl polyester systems, first, we focused on poly(ethylene terephthalate) (PET). A consistent phenomenon was observed after conducting tests on PET under various crystallization conditions (covering from T_c_ = 210 °C to 225 °C) after 2 min of melting at 280 °C; an irregular ring-banded morphology was formed with “positive”-type spherulites. Owing to PET’s irregularly banded morphology, further characterization was conducted by using SEM to expose the surface morphology pattern. Figure 1 displays the irregular bands of PET film at T_c_ = 220 °C, which is composed of a fibrillar-stripe morphology with a zig-zag array. This fibrillar morphology spreads outward from the center of the nucleus toward the periphery radially. It is apparent from the fibrillar pattern that there is no distinguishable contrast between the ridges and valleys. For comparison, in Appendix A shows POM images for PET crystallized at a broad range of T_c_ = 200, 205, 210, 215, 220, and 230 °C.

In addition to the POM results, PET’s iridescent properties were also depicted by measuring a series of PET film samples crystallized at T_c_ = 210–225 °C. Figure 2 shows POM images and the iridescent test results for crystallized PET films. The lack of iridescent coloration from the PET films is well correlated with the PET’s disordered ring bands and irregularity of the interior lamellae. Since the PET’s surface morphology exhibits random orientations, there is no possibility of photonic iridescence, as demonstrated by the absence of coloration in the photo images of PET. When light reaches the surface’s irregular topography of crystallized PET film, it reflects and propagates in all random directions without producing any constructive light interference coloration spectra. 

By comparing the crystal morphology, surface relief pattern, interior lamellar assembly, and iridescent properties in each of the periodic patterns of various aryl polyesters, it is clear that only PET is not capable of producing structural coloration. This is consistent with numerous publications [34,35] which have stated that neat PET cannot form regular ring-banded morphology under various thermal conditions as well as the orderly arrays from the top and interior surfaces. The remaining four polyesters, i.e., PTT, POT, PNT, and PDT, can form ring-banded spherulites with their unique properties. The top surfaces and internal crystal plate assemblies were thoroughly analyzed, and the results indicated that the aforementioned polyesters could form hierarchical structures with ordered lamellar arrays.

Figure 3 shows the SEM images for crystallized PTT films in a wide range of temperatures at T_c_ = from 90 °C to 175 °C. Ringless PTT spherulites are formed only at T_c_ < 90 °C (Figure 3a), and their surfaces are relatively smooth and lack an up-and-down topology compared to the regularly banded spherulites. Figure 3b demonstrates the marginal ring-banded morphology in PTT, which, nevertheless, creates highly organized structures in small-radius spherulites. The undulated morphology can be observed, which acts as a grating surface, resulting in a soft-hue iridescent color. Figure 3c clearly illustrates the ring-banded structures at T_c_ = 150 °C, where the distinct lamellae in the ridge and valley zones act as gratings to generate vivid structural coloration. With band spacing at ca. ~5.5 µm, interference can take place optimally and produce intense colors. Ring-banded spherulites with a wider band spacing (ca. ~40 µm) are formed at T_c_ = 175 °C, as shown in Figure 3d, and are unable to support sufficient interference. Furthermore, compared to the previous two ring-banded morphologies, there is a lower difference in elevation between the ridge and valley, leading to the absence of interference in this condition. 

As earlier interpreted by Lugito et al. [56], Figure 4 illustrates the top surface and interior SEM morphology of ring-banded spherulites of crystallized PTT films, using the same thermal treatments as in the present study to correlate with neat PTT with photonic iridescence. Highly organized ring-banded PTT spherulites crystallized at T_c_ = 165 °C are generated from the top and interior surfaces, which may periodically form plate by plate. The crystal lamellae beneath the ridges are stacked vertically, while those beneath the valleys are stacked horizontally. The morphology of this structure resembles those in some bio-iridescent structures. The orderly arrays on the top surface, as well as the internal microstructures with layered assembly, collectively attribute to the intriguing photonic features in the banded PTT.

By comparison, PET and PTT differ only by one methylene (-CH_2_-) in their chemical repeat units; however, PET hardly forms periodic ring bands with highly corrupted patterns at most, while PTT easily forms regular and well-defined ring-banded spherulites within a wide range of T_c_ (ca. 100–170 °C), as discussed above. Note that both PBT and PHT, with methylene segment units of 4 and 6, respectively, do not form ring bands at all when crystallized at any feasible T_c_s. Poly(pentamethylene terephthalate) (PPentaT) or poly(heptamethylene terephthalate) (PHepT), similar to PTT, also is reported to form ring bands, although the banded patterns of PPentaT and PHepT tend to be less ordered than does that of PTT. By examining the chemical structures of the repeat units, inquirers may ask with curiosity: “Do arylate polyesters with an even number of methylene segments always display ringless spherulites; oppositely, are arylate polyesters with odd number of methylene segments always capable of forming ring-banded spherulites (within a suitable T_c_ window)?” Superficially, the rule holds for homologous series from PET to PHT; however, with the number of methylene segments reaching beyond 8, the rule is no longer valid, that is, the even–odd rule holds only partially and not universally. 

### 2.2. Iridescent Tests for Aryl Polyesters

Some subscales of a hierarchical structure can act as holographic gratings when exposed to light, revealing the interference pattern as a result of the periodic assembly. Periodically banded crystals have ordered microstructures arranged in grating-like arrays that can interact constructively with some light wavelengths but not with others to produce spectral coloration. Figure 5 illustrates the interference between ring-banded morphology and irregular ring-banded morphology to generate color spectra. Figure 5a depicts arylate polyesters with regular ring-banded spherulitic structures when viewed from the top surface in conjunction with light. Simultaneously, incident light is reflected regularly and forms waves, as in Figure 5b, when viewed from the side. The mechanism of light interference can be explained by the fact that most of the sources produce white light waves that randomly travel in all directions. The situation means that light wavelengths emitting from a source do not have a constant amplitude, frequency, or phase. Moreover, once they reach the ridge and valley, they create an interference pattern that can be tuned to create color. The incident light splits into numerous beams that travel in different directions and generate constructive and destructive interference. Interference is the process by which two waves combine to form a combined wave of different or identical amplitude. By contrast, when PET crystallizes at a specific temperature range, it does not form the regular ring-banded morphology but rather an irregular ring-banded morphology reminiscent of fibrillar lamellae. Additionally, non-periodic disorderly microstructures were employed in the irregular banded and ringless structure, which do not support photonic reflection. No color interference is revealed because reflected light can travel randomly in any direction. The opposite situation from the previous phenomenon is illustrated in Figure 5c, where light is reflected in all directions when it strikes the surface of irregular ring-banded spherulites. The separation of the distance between the ridge and valley is too far in this case, resulting in no interference in this situation. This is related to the random reflection phenomenon. Moreover, as illustrated in Figure 5d, no coloration is produced by reflection from the irregular PET’s surfaces. Thus, a similar mechanism to nature’s photonic structures, the hierarchical grating and periodic orderly structure in polymers has a unique effect on iridescence, and the structural coloration takes this microstructure into account.

Poly(trimethylene terephthalate) (PTT) is an aryl polyester with multiple types of ring-banded spherulites. Lugito et al. [35] investigated the morphology of neat PTT at a T_c_ = 165 °C. Additionally, this study employs an iridescent properties analysis to determine the correlation between the top surface of neat PTT ring-banded spherulites and their inner architecture in greater detail. Figure 6 shows the POM morphology of neat PTT at a wide range of T_c_ = 90–190 °C. As neat PTT is crystallized at temperatures from T_c_ = 90 °C to 190 °C, the morphology of PTT spherulites begins with a ringless pattern, and then gradually transitions to ring-banded spherulites with a marginal band regularity at T_c_ (100–150 °C); it exhibits a regular ring-banded one at intermediate T_c_ (150–175 °C), and finally a corrupted pattern with radial lamellar splash at a very high T_c_ (>175 °C). According to Hong et al. [57], the results are consistent with their finding that PTT crystallizes in three distinct kinetic regimes depending on the crystallization temperatures. In regimes I, II, and III, the morphology of spherulites changes from a straight axialite to a circular-banded morphology, and then to a ringless morphology, respectively, as the temperature drops. 

Next, the iridescent properties of the periodically banded PTT crystals (at T_c_ = 90–190 ℃) were tested using in-house setups. Figure 7 shows representative iridescent patterns of neat PTT films, where, in Figure 7a,b, it can be observed that the morphology of two ringless PET spherulites at 90 °C or 100 °C that do not produce photonic reflection due to the lack of an orderly structure. By contrast, for the inter-band spacing = 4 µm–15 µm, the interference coloration becomes the most intensely visible (Figure 7c–f) for PTT films crystallized at T_c_ = 140, 150, 160, and 165 °C, respectively. The iridescent properties are only supported by a ring-banded structural morphology. On the one hand, the color variations are observed by starting with a soft hue color for PTT with a borderline banding morphology, which can intensify upon refinement of the band regularity. In order to magnify this effect, the inter-band spacing of the PTT spherulites were custom adjusted by varying the T_c_. On the other hand, the regular ring-banded morphology with large inter-band spacing (ca. ~40 µm) at T_c_ = 175 °C does not produce iridescence (Figure 7g). Likewise, for the ringless PTT spherulites packed with radial lamellae splashing from a central nucleus, the absence of an orderly structure makes it unable to produce iridescent coloration (Figure 7h).

### 2.3. Periodic Microstructures and Iridescent Features of POT and PNT

The banded patterns in PPenT or PHepT (5 and 7 methylene segments between terephthalates, respectively, in chemical repeat units) tend to be less ordered; therefore, this work did not aim to include these two for evaluation. Similar to PTT, both POT and PNT display orderly rings when crystallized at suitable ranges of temperature. POT films were isothermally crystallized by quenching from a maximum melt temperature (T_max_) = 160 °C and isothermally held within a range of T_c_ varying from 85 °C to 115 °C. An earlier work probed the assembly from nanocrystals in hierarchical levels to final periodically aggregated spherulites of POT [58], which has revealed that the interior surface of a banded POT spherulite is composed of crystal-stacked shell-like piles aligned radially along the ridges and nuclei, which, in turn, are surrounded by featureless and flat lamellae in the valleys. At the ridge, these lamellae grow and periodically branch outward along the radial direction, and then re-assemble in the valley along the tangential direction, forming a periodic hierarchical structure with repetitive cycles. The topology pattern alone was not sufficient to interpret the assembly in the entire bulk; thus, an analysis was necessary for the interior structural assembly of neat POT (fractured interiors in multifacet 3D views)**.** Like nacre, it is not only the top-surface morphology but also the inner layer micro-/nanoscale grating assembly that is responsible for the pearl-like iridescence. Huang et al. [58] investigated and reported the interior morphology of neat POT periodic bands at T_c_ = 105 °C, which happens to be the most regular pattern of the ring-banded morphology in the T_c_ range from 85 to 125 °C. The interiors of POT spherulites is composed of crystal-stack piles self-aligned radially along the ridges and nuclei, which, in turn, are surrounded by featureless and flat lamellae in the valleys. At the ridges, these lamellae grow and branch outward in a radial direction, and then re-assemble in the valleys along the tangential direction, forming a periodic hierarchical structure. 

The interior assembly of POT is further analyzed and expounded in this work using similar techniques. To further expound the assembly in greater details, the correlation between the top-banding pattern and interior structure of neat POT is depicted in Figure 8a,b for the top-surface rings and fractured interiors neatly packed with orderly layers, respectively. White arrows in the figure are used to mark the alternate lamellar orientations with perpendicular intersections and interfaces between the onion-like layers. This optimum height difference between the periodic ridge and valley produces substantial light interference under these conditions. As the temperature increases, the POT spherulites’ patterns change systematically. Between crystallization temperatures of 85 °C and 105 °C, distinct ring-banded spherulites with assembly order are visible. It appears that the radii of the spherulites and inter-band spacing increase proportionately with increases in T_c_. In summary, POT has a grating-like architecture, and the top surfaces and internal lamellar arrangements of banded POT are responsible for the occurrence of periodic changes in the optical interference—viewed as alternate colored circular rings with a lambda-tint plate in POM [58].

The iridescence of orderly banded vs. ringless POT films (crystallized at two different temperatures) was doubly tested and compared, with experimental setups similar to those used in a previous work [35]. In Figure 9a,b, it can be observed that there are POM micrographs of POT spherulites with orderly ring bands vs. ringless patterns, respectively, at T_c_ = 90 and 115 °C. The focus is placed on an in-depth analysis of the correlations among the top surface, internal crystal assembly, and the iridescent properties formed by POT’s crystallized films at specific isothermal temperatures. An additional photonic iridescent analysis reveals that POT films with highly ordered structures are capable of effective interference with white light to produce prominent iridescent coloration, as illustrated in Figure 9a1,b1. The coloration begins with a soft-hue color for POT at 85 °C with increasing intensity up to a temperature of 105 °C due to POT’s most regularly ring-banded morphology. The size of the spherulite may affect its interior microstructures, and thus the iridescent characteristics produced by the periodically assembled POT. By contrast, non-iridescence for POT films (mostly ringless patterns) is seen at T_c_ = 105–120 °C (Figure 9(b1)). Above the temperature of T_c_ = 105 °C, the ring bands of POT spherulites become less orderly and the iridescent characteristics are reduced and eventually cease to exist for ringless or highly corrupted patterns.

PNT, an aryl polyester containing nine-fold methylene (CH_2_) segments between successive terephthalates, was also probed for correlations between the nano- or microstructures and their iridescent capacities. Tu et al. [59] investigated the morphology of PNT spherulites by employing SEM for correlating the top surface topography with the interior crystal arrangement to determine its self-assembly mechanism. In order to utilize the PNT spherulites for potential applications, this study promoted the photonic reflection experiments of the sample to investigate the iridescent properties of PNT spherulites. First, the PNT samples were prepared by melting at a melting temperature (T_max_) of 120 °C. Within a specific crystallization temperature (T_c_) range, PNT is also packed into periodic ring-banded spherulites. The PNT crystallizes at temperatures ranging from 55 °C to 85 °C, and all exhibit a similar ring-banded morphology, differing in the inter-band spacing. 

Subsequently, photonic reflection was observed on neat PNT as shown in Figure 10a–f, displaying the iridescent features of PNT films at T_c_ = 55, 65, 70, 75, 80, and 85 °C, respectively. The coloration intensity begins as a soft hue and gradually increases as the crystallization temperature increases. The most intense color occurs in PNT crystallized from 70 °C to 85 °C. From the PNT-banded spherulites, these periodic hierarchical lamellae with perpendicular orientations create an orderly microstructure suitable for interference with light. The similar regularity of these two types of banded patterns facilitates the iridescent properties in PNT crystals [59]. 

Similar to the work on PTT or other polyesters, an earlier analysis of the top surface and internal analysis perspectives of neat PDT (10 methylene segments in chemical repeat unit) crystallized at T_c_ = 80–95 °C has been described by Yang et al. [60]. The ring-banded PDT spherulites are composed of cyclic patterns of a protruded ridge and a flat-plane untextured valley area, as viewed from the top surface. The ridge’s internal crystal plates are packed with normal-oriented lamellae and their branches at an angle, whereas in the valley zone, the crystal plates are arranged horizontally (to the substrate plane). Naturally, while corrupted ring bands in PDT films at T_c_ = 105–115 °C produce non-iridescence, the regularly banded PDT films crystallized at T_c_ = 80–95 °C are capable of displaying iridescence, as proven in a concurrent earlier work [60]. 

As a brief summary, the morphologies, iridescent properties, laser diffraction patterns, and grating architectures of five aryl polyesters (PET, PTT, POT, PNT, and PDT) are collected and compared in Figure 11a–c, which display the POM graphs, iridescence, laser-light diffraction, respectively. By comparing each of the aryl polyester’s characteristics that produce various levels of intensity of structural coloration spectra, this section selects representatives of each type of aryl polyester with the most optimal iridescence. The laser diffraction phenomenon was also further investigated. For the irregularly (or corrupted) ring-banded spherulites, on the one hand, the lack of orderly structures results in the absence of diffraction patterns upon interacting with the laser light beam (Row I). On the other hand, smooth, circular diffraction patterns emerge in response to the regularly ring-banded morphologies, where multiple orders of circular diffraction patterns are observed in the remaining four arylate polyesters (Rows II–V).

Table 1 shows the laser-light diffraction results for four aryl polyesters. The calculated result is consistent with the phenomenon that the d-spacing produced by laser diffraction is approximately equal to the inter-band spacing of banded aryl polyesters as determined by POM and SEM on top-surface bands and in dissected interiors. The purpose of the laser diffraction analysis was to demonstrate the effectiveness of this technique for identifying well-organized and periodically banded structures with ring-banded periodicity in the polymers’ periodic crystalline aggregates. 

Arylate polyesters with longer methylene segments also exhibit the same characteristics of ring bands in their crystallized films in respectively suitable temperature ranges. As an example, an arylate polyester with 12 methylene segments in the chemical repeat unit, poly(dodecamethylene terephthalate) (PDodT or P12T), is used as a model for comparison. Again, the interior assembly of PDodT is further expounded in this work using the techniques in an earlier investigation [39]. Figure 12 shows both the POM micrograph and the interior-fractured SEM image for P12T (PDodT) films crystallized at T_c_ = 90 °C. The banding order at T_c_ = 90 °C is similar to the result in an earlier work on P12T at T_c_ = 96 °C. The fractured interior apparently displays a periodic grating assembly and inter-band discontinuous interfaces, where onion-like alternate strut-rib shells are seen to surround the nucleus center. The shell thickness (5.5 μm) of this onion-like structure in the SEM evidence matches perfectly with the optical inter-band spacing as revealed in the POM image, which suggests that the discontinuous (with detached interfaces) but otherwise orderly layered gratings are responsible for the optical rings of alternate birefringence changes.

Universal assembly features in the homologous arylate polyesters are summarized here. The as-discussed SEM interior dissection of the banded crystal aggregates of several homologous polyesters have revealed two contrasting birefringence colors from two species of mutually intersecting lamellae. Such universally common features of grating assemblies and correlations with the top surfaces and interiors can be summed up as schematics in Figure 13 for PTT, POT, PNT and PDT with methylene segments = 3, 8, 9, and 10, respectively, per chemical repeat unit.

In comparison to the discussed polymers, inorganic crystalline minerals such as moonstones, opals or gems, or biological organic species in nature have been known to abundantly utilize micro/nanostructures for producing structural coloration [61,62,63,64]. The known photonic-related micro/nanostructures, in nature, are mainly based on periodic gratings or orderly assembled microspheres, but never from continuously helix-twist crystal plates. This work surveyed several arylate polyesters that are capable of forming periodic aggregated crystals/spherulites with periodic gratings. Upon self-assembly into periodical aggregation, they share similar orderly grating assemblies, resembling those in nature’s iridescent structural coloration; thus, they display similar iridescent patterns, though not as versatile. The above universal features in the arylate polyesters demonstrate that the periodic bands are constructed by a similar assembly mechanism that originates from common habits of materials’ crystallization. The circular patterns from the laser-light diffraction means the characteristic of the ring-banded spherulites is uniform and regularly distributed. The crystallization of polymers has advanced a novel approach for mimicking the scales of nature’s bio-photonic crystals. Thus, recent rapid advancements in nanotechnology demonstrate that it is possible to mimic even the most intricate structures found in nature by using custom-made arylate-polymeric spherulites. As a result, structural coloration has been extended to synthetic polymers, as optical and physical investigations of their mechanisms of crystal assembly would be immediately confirmed through fabrication of reproduction. 

## 3. Materials and Methods

Several arylate polyesters used in this work are summarized in Table 2. 

PET and PTT were dissolved in dichloroacetic acid (C_2_H_2_Cl_2_O_2_) with concentrations of 4 wt.% and 2 wt.%, respectively. The sample was first drip-cast on a glass substrate at 80 °C as a thin film, and then, it was put into a vacuum oven for 24 h to volatilize the residual solvent before heat treatments for crystallization and analysis. For thermal treatment/annealing, PET samples were heated to melt on a hot plate at T_max_ = 280 °C for 2 min to erase any prior crystal or thermal histories before being rapidly replaced to another hot stage preset at a designated isothermal T_c_ = 205–220 °C. Neat PTT film samples were heated on a hot plate to T_max_ = 260 °C for 5 min to erase the prior crystals or thermal histories, and then, they were rapidly replaced to another hot stage preset at a designated isothermal T_c_ = 90–190 °C. For other arylate polyesters with long methylene segments in chemical repeat units, solvents of lower polarity are sufficient to dissolve. Chloroform was used as the good solvent for POT, PNT, and PDT systems. POT and PDT films were configured with casting from 2 to 4 wt.% of the solution concentration, while PDT was configured with 1–2 wt.% solution concentration. The polymer solutions were drip-cast on the hot-plate preset at 35 °C and waited until the solvent was fully evaporated to form a solid film. The thin films on the glass slide were left for one day to allow the solvent to evaporate completely, and finally vacuum-dried for 1–2 days. Prior to characterization, the dried film samples were heated to T_max_ = 160 °C for POT, T_max_ = 120 °C for PNT, and T_max_ = 165 °C, for 1–2 min to erase the thermal history and then, rapidly transferred to the hot stage preset at various T_c_ values until full crystallization. Premelting of molten polymers at T_max_ above their respective T_m_ was for erasing the prior thermal histories and modulating the nuclei density.

### Apparatus

Polarized-light optical microscopy (POM) (Nikon Optiphot-2, POM, Tokyo, Japan), equipped with a Nikon Digital Sight (DS)-U1 camera control system and a microscopic hot stage (Linkam THMS-600 with T95 temperature programmer), was used to characterize the crystalline morphology of the polymers. Furthermore, objective lenses with magnifications of 4×, 10×, 40×, and 100× were used to observe and capture the images at various magnifications. The 100× objective lens required oil contact with specimen, and was used by placing a drop of oil for the microscope oil-contact lens on the back of the sample glass slide. 

High-resolution field emission scanning electron microscopy (HR-FESEM) was used to reveal the lamellar structure in the top surface and cross-section interiors of the crystallized samples. Samples were examined and characterized using high-resolution field emission scanning electron microscopy (Hitachi SU8010, HR-FESEM, Tokyo, Japan). The crystallized films (flat and uniform on glass substrate) were carefully precut with a diamond knife on the back of the glass slide and fractured in a low-temperature liquid nitrogen environment to prevent unevenness of the fractured section and avoid the polymer’s ductility interfering with subsequent observations. Some specimens might have to be etched with proper solvents, which were dried prior to sputter coating. Then, the etched sample was positioned on the surface of an aluminum stand using a carbon glue to adhere the sample to the stand. All samples were sputter-coated with gold-vapor deposition prior to SEM characterization. 

For the photonic iridescent measurements, the samples for photonic reflection tests were prepared by solution-casting the polymers on a 1.8 × 1.8 cm^2^ micro-glass coverslip, and crystallized at specific T_c_ values to produce the desired ring patterns. Iridescent observations were taken using an in-house-made photo-reflection setup. The crystallized sample was placed in a fixed position on black background and exposed to visible white light (LED sources), while the iridescent phenomena were captured with camera images.

The laser diffraction measurements were carried out with a green laser light (model no. PM 851) with a wavelength (λ) of 532 ± 10 nm. A crystallized polymer sample was placed between the laser light and screen, where the regularly circular rings of the polymer spherulites acted as optical gratings for light diffraction. The band spacing values (d) of the crystallized spherulites in the film specimens were estimated from the light diffraction results using the Bragg’s equation: nλ = 2d*sin(θ); θ = tan^−1^ (r/L)(1)

## 4. Conclusions

Five members of homologous aryl polyesters are used for demonstrating the universal features in morphological interpretations of periodically ring-banded spherulites, and the orderliness of the grating microstructures in periodicity are checked and confirmed with iridescent tests. PET, which is incapable of producing regular ring-banded morphology, cannot produce any interference, and thus no iridescent properties. Other than PET, the other members of aryl polyesters can form ring-banded spherulites when crystallized at a specific window of temperatures. PTT crystallized at a specific temperature can form regularly ring-banded spherulites; correspondingly, PTT-banded spherulitic aggregates would exhibit distinct interference with bright coloration iridescence. PNT also displays regular bands when crystallized at a suitable T_c_, and the regularly banded PNT crystals are also capable of producing vibrant coloration spectra upon interfering with white light. By contrast, both POT and PDT can also be regularly banded; however, they have smaller sizes of the spherulites’ radii than those of PTT, and the small sizes and less orderly assemblies both result in a softer hue of less intensity or absence of iridescent properties. 

The characteristic micro/nanostructures in an orderly assembly of periodic bands from the investigated aryl polyesters enable them to display iridescent properties of different levels of intensity that are dependent on the regularity of the microstructures. Such hierarchical structures of banded polymers display a unique effect on structural coloration. Circular diffraction patterns with laser light give a clue to double-check the regular periodicity in the aryl polyesters’ ring-banded spherulites. This additional evidence provided proper support for the corrugate-board grating morphology of the arylate polyesters displaying periodic bands when viewed in POM. Finally, it can be concluded from these in-depth analyses on five different aryl polyesters that the ring-banded spherulites are periodically grating-assembled, and they are also capable of displaying light interference when the periodicity and micro/nanostructures are properly tailor-made. The ring-banded morphology creates a complex corrugated grating morphology, which can be packed into periodic microstructure resembling grating-assembled platelets on the biostructure’s surface that perform the function of light interference leading to iridescent coloration.

## Figures and Tables

**Figure 1 ijms-24-15538-f001:**
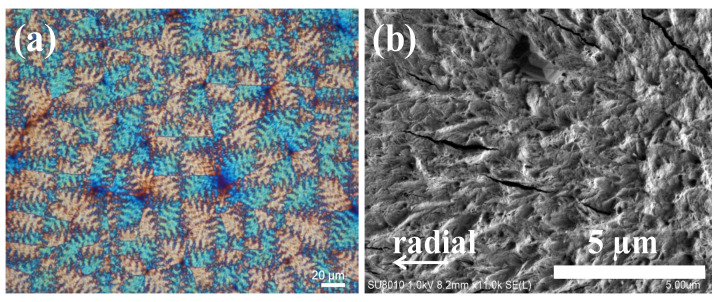
PET crystallized at T_c_ = 220 °C after being melted for 2 min at max-melt temperature (T_max_) 280 °C: (**a**) POM image and (**b**) SEM graph for top surface.

**Figure 2 ijms-24-15538-f002:**
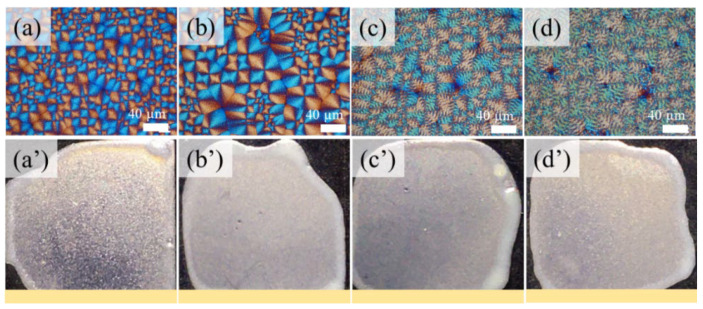
POM graphs and weak or non-iridescence of PET films crystallized at different crystallization temperatures (T_c_): (**a**,**a′**) 210 °C; (**b**,**b′**) 215 °C; (**c**,**c′**) 220 °C; (**d**,**d′**) 225 °C, after being melted at T_max_ = 280 °C for 2 min (scale bar = 40 μm).

**Figure 3 ijms-24-15538-f003:**
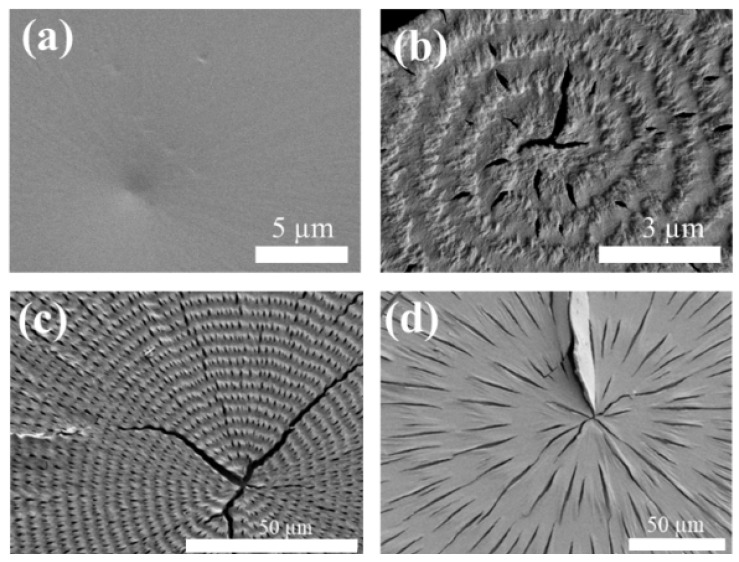
SEM images of crystallized PTT films at various crystallization temperature (T_c_) values: (**a**) 90 °C; (**b**) 100 °C; (**c**) 150 °C; (**d**) 175 °C.

**Figure 4 ijms-24-15538-f004:**
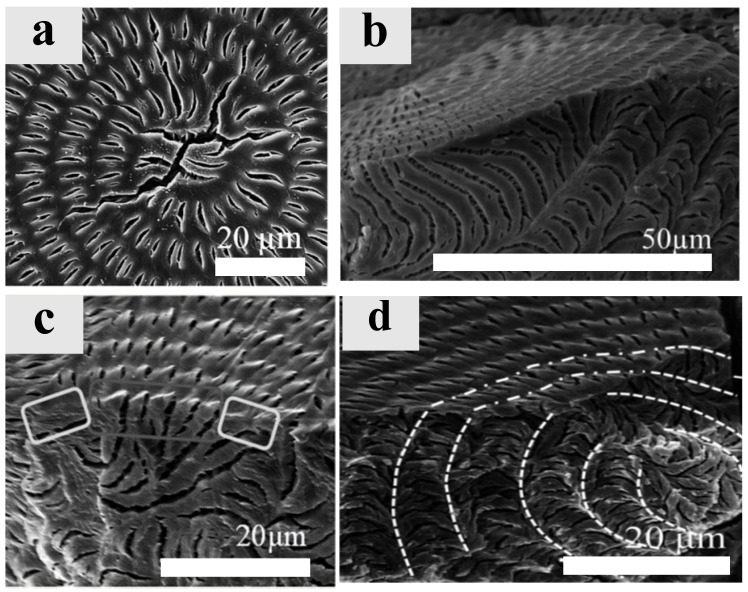
SEM micrographs of (**a**) surface and (**b**–**d**) the interior assembly of neat PTT crystallized at T_c_ = 165 °C, after being melted at T_max_ = 260 °C for 5 min [56] (reproduced with 2023 copyright permission by Woo).

**Figure 5 ijms-24-15538-f005:**
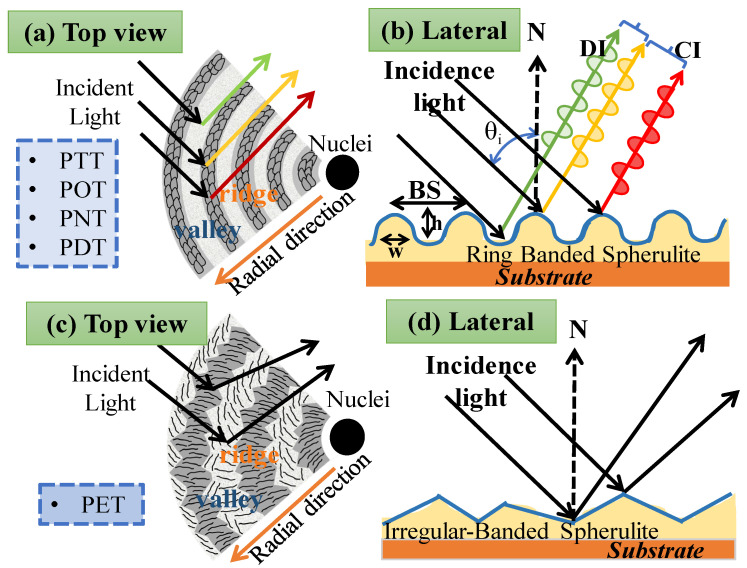
Schematics of light interference on orderly vs. disorderly assembly: (**a**) Top view of aryl polyesters with regular ring-banded spherulites; (**b**) side view of aryl polyesters with regular ring-banded spherulites; (**c**) top view of aryl polyesters with corrupted-banded spherulites; (**d**) side view of aryl polyesters with corrupted-banded spherulites. N, normal direction; BS, band spacing; w, width; h, height profile; θ_i_, angle of incidence; DI, destructive interference; CI, constructive interference.

**Figure 6 ijms-24-15538-f006:**
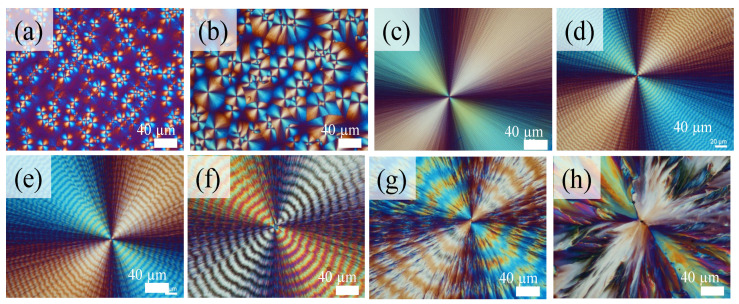
POM graphs of neat PTT crystallized at different crystallization temperature (T_c_) values: (**a**) 90 °C; (**b**) 100 °C; (**c**) 140 °C; (**d**) 150 °C; (**e**) 160 °C; (**f**) 165 °C; (**g**) 175 °C; (**h**) 190 °C, after crystallized at T_max_ = 260 °C for 5 min (scale bar = 40 μm).

**Figure 7 ijms-24-15538-f007:**
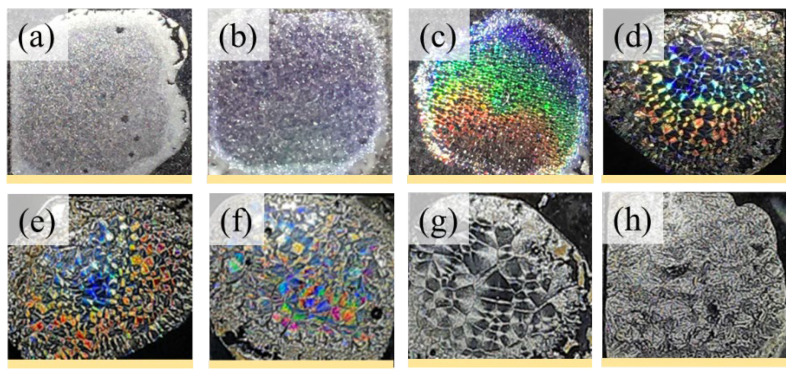
Iridescent coloration changes of banded PTT in films crystallized at various temperatures (T_c_) values: (**a**) 90 °C; (**b**) 100 °C; (**c**) 140 °C; (**d**) 150 °C; (**e**) 160 °C; (**f**) 165 °C; (**g**) 175 °C; (**h**) 190 °C (yellow scale bar = 1.8 cm).

**Figure 8 ijms-24-15538-f008:**
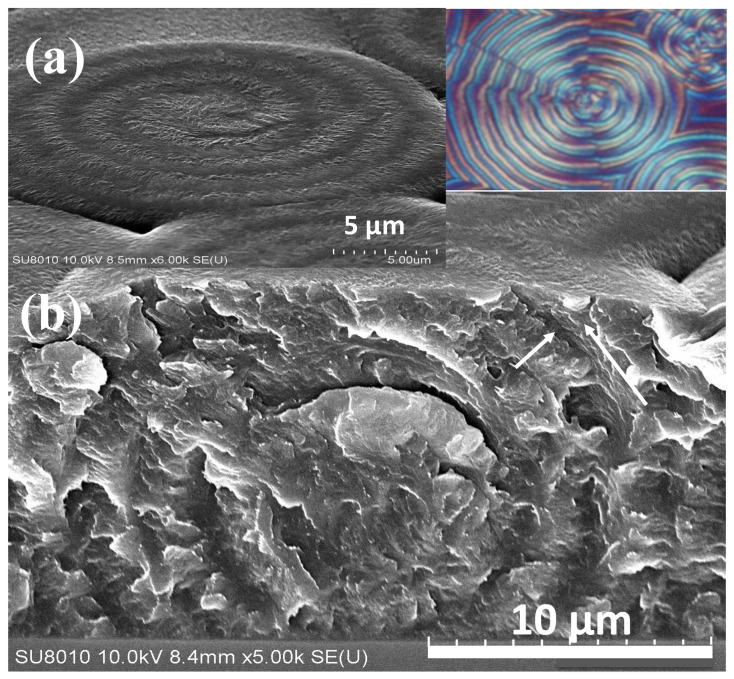
SEM micrograph (inset upper right for POM) for (**a**) the top surface with spiral-spin rings originating from a nucleus center and (**b**) the interior banded POT crystals at T_c_ = 105 °C, fracture across the nucleus, exposing grating onion-like spheroid layers, the white arrow revealing radial and tangential orientation.

**Figure 9 ijms-24-15538-f009:**
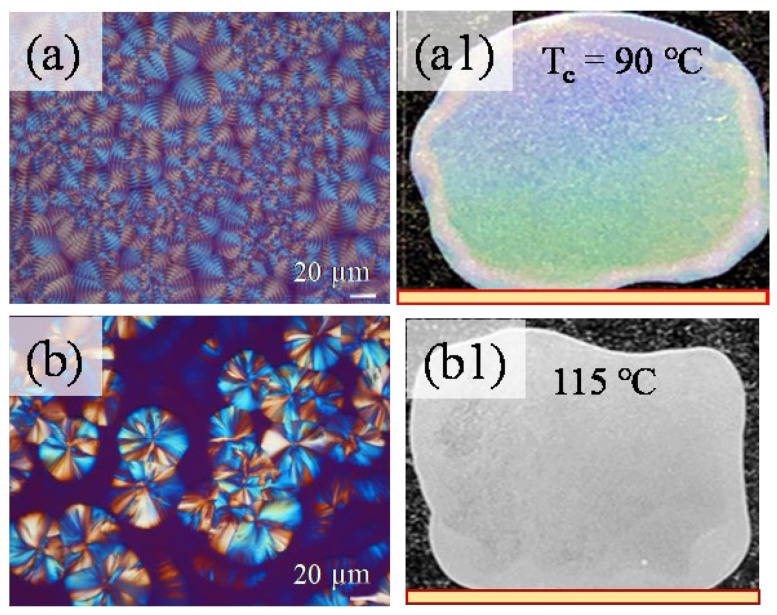
(**a**,**b**) POM band patterns with (**a1**,**b1**) iridescence in orderly vs. corrupted bands in POT films crystallized at T_c_ = 90 vs. 115 °C, respectively. Scale bar in POM = 20 μm. Yellow scale bar in images = 18 mm.

**Figure 10 ijms-24-15538-f010:**
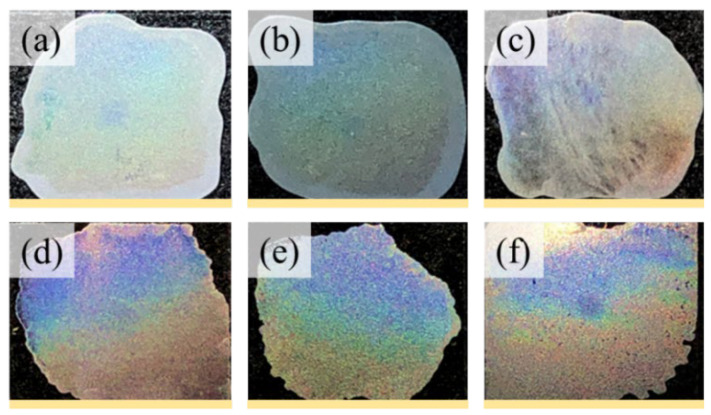
Photonic iridescence of neat PNT films, all with orderly ring bands, crystallized at different crystallization temperature (T_c_) values: (**a**) 55 °C; (**b**) 65 °C; (**c**) 70 °C; (**d**) 75 °C; (**e**) 80 °C; (**f**) 85 °C, after being melted at T_max_ = 120 °C for 2 min (scale bar = 1.8 cm).

**Figure 11 ijms-24-15538-f011:**
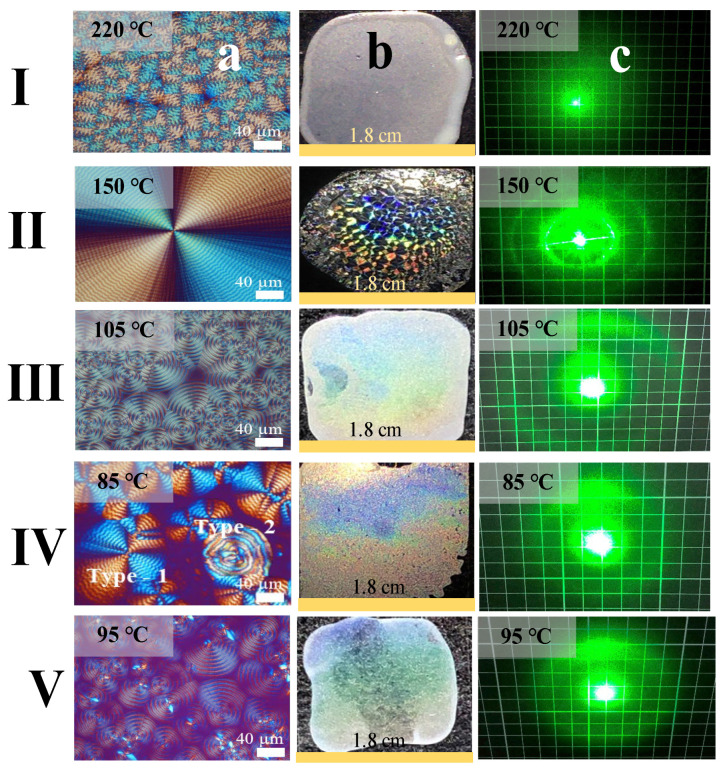
Comparison of POM graphs, iridescent photos, laser diffraction of several homologous aryl polyesters, Columns: (**a**) POM; (**b**) iridescent properties; (**c**) laser diffraction, in Rows: (**I**) PET; (**II**) PTT; (**III**) POT; (**IV**) PNT; (**V**) PDT.

**Figure 12 ijms-24-15538-f012:**
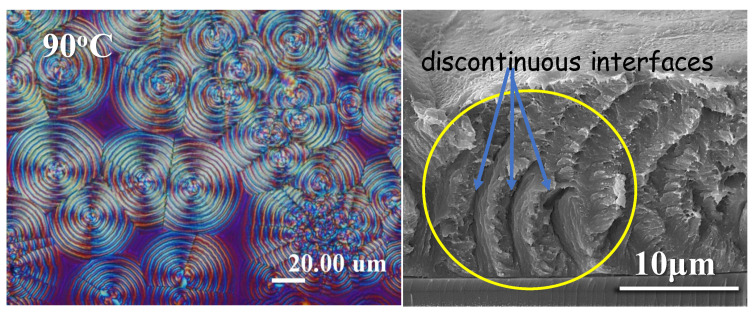
POM and interior-fractured SEM images for P12T (PDodT) spherulites in films crystallized at T_c_ =90 °C, yellow circle displaying periodic grating assembly and discontinuous interfaces and onion-like alternate strut-rib shells (film thickness 1~3 μm, cast from 1 wt.% solution).

**Figure 13 ijms-24-15538-f013:**
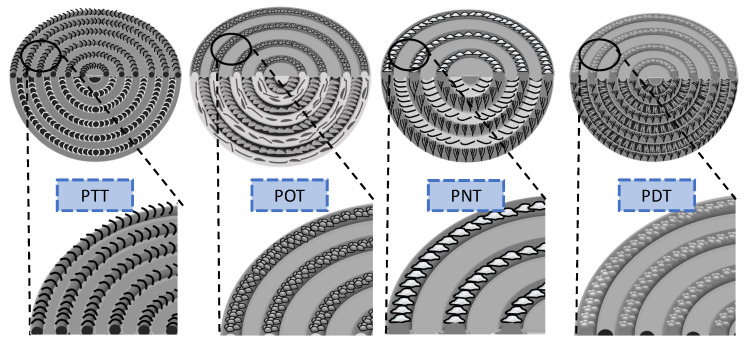
Schematic illustrations showing the commonality of corrugated-grating morphology in four different aryl polyesters (PTT, POT, PNT, and PDT).

**Table 1 ijms-24-15538-t001:** Laser diffraction results, showing the correlation among the band patterns of arylate-polymeric spherulites.

Polymer	d-Spacing (Laser Diffraction) (µm)	BS (POM) (µm)	BS (SEM for Top Surface) (µm)	BS (SEM for Interior) (µm)	θ1 (°)	θ2 (°)
PTT	5.5	5.5	5.5	-	5.7	11.2
POT	3.6	3.6	3.7	3.6	17.1	-
PNT	7.1	6.9	7.1	7.1	8.7	-
PDT	3.1	3.3	3.2	3.1	12.1	-

BS—band spacing.

**Table 2 ijms-24-15538-t002:** Chemical structures and physical properties of polymer materials.

Polymers	Chemical Structures
Poly(ethylene terephthalate) (PET)	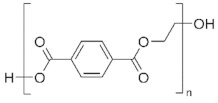
Poly(trimethylene terephthalate) (PTT)	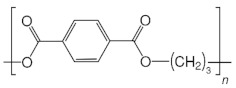
Poly(octamethylene terephthalate) (POT)	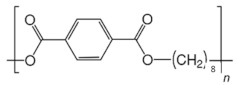
Poly(nonamethylene terephthalate) (PNT)	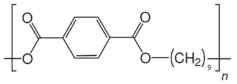
Poly(decamethylene terephthalate) (PDT)	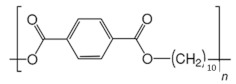

## Data Availability

The data presented in this study are available in article.

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
