# Peer review of "Iridescent Features Correlating with Periodic Assemblies in Custom-Crystallized Arylate Polyesters"

_ijms, 2023, doi:10.3390/ijms242115538_

Round 1
Reviewer 1 Report
The reviewer read the manuscript entitled "Iridescence Features Correlating with Periodic Assemblies in Custom-Crystallized Arylate Polyesters". The manuscript has a new concept of polymer and material science. By the surface control of spherulites, the authors could observe the structural color.
1. The authors could control the pitch of banded crystals. How about the annealing temperature dependence of the structural color?
2. How about the material dependence of structural color?
3. The authors should evaluate these data more quantitatively. The authors should carry out the UV-Vis spectra and/or ATR-IR spectra.
Reviewer 2 Report
In this manuscript, five different aryl polyesters, including PET, PTT, POT, PNT, and PDT, exhibit ring-banded spherulites when crystallized at suitable temperatures. These spherulites display periodic birefringence patterns reminiscent of iridescent biological structures, with PET being the only exception, incapable of producing such morphology and color iridescence. The manuscript is well written, but the authors must address some concerns before it is considered for publication.
1) Could you add additional information regarding the practical application of iridescence properties?
2) Several of the references cited in the introduction section are dated. Could you consider incorporating more recent references for enhanced currency?
3) A Typo in Line 138 and 143, degree Celsius is "°C" instead of "oC". For example, °C in Line 133 is correct.
4) I like the beautiful figures in this manuscript!
